# GenVOG-DiT: A Transformer-Based Diffusion Model for Pose-Driven, Patient-Agnostic Nystagmus VOG Video Generation

**Aimon Rahman**[1]                                                    ARAHMA30@JHU.EDU
[1] *Johns Hopkins University, Department of Electrical and Computer Engineering, Baltimore, MD, USA*

**Kemar E. Green**[2]                                                  KGREEN66@JHMI.EDU
[2] *Johns Hopkins University, Department of Neurology, Baltimore, MD, USA*

**Vishal M. Patel**[1]                                                 VPATEL36@JHU.EDU

**Editors:** Accepted for publication at MIDL 2026

## Abstract

Nystagmus, an involuntary eye movement indicative of neurological and vestibular disorders, is traditionally diagnosed using costly equipment or expert visual inspection: both of which limit accessibility in nonspecialist settings. Recent advances in computer vision and deep learning present an opportunity to automate the detection of nystagmus from standard video recordings. However, progress is hindered by the scarcity of publicly available video datasets due to privacy concerns surrounding ocular biometric data. In this work, we propose the use of synthetically generated eye movement videos to mitigate data limitations. Using video diffusion models, we simulate diverse clinically plausible nystagmus patterns without relying on real patient data, enabling scalable training while preserving privacy. We show that models trained on synthetic data generalize effectively to real-world settings and show potential for integration into telehealth applications. Our approach advances the development of accessible, generalizable, and privacy-aware diagnostic tools for eye movement disorders.

**Keywords:** Video Generation, Diffusion Transformer, Nystagmus.

## 1. Introduction

Nystagmus is an involuntary rhythmic oscillation of the eyes characterized by a slow drift of the eye in one direction followed by a corrective movement in the opposite direction (Wagle et al., 2022b; Leigh and Zee, 2015; Anastasio et al., 2022). It can occur in various planes: horizontal, vertical, or torsional, and typically presents in one of two waveform patterns: jerk nystagmus, where the eyes drift slowly in one direction, then jerk back quickly, and pendular nystagmus, where the eyes swing back and forth in a steady, pendulum-like motion without a distinct fast phase. Although the direction, velocity, and pattern of the nystagmus can localize dizziness to a peripheral vestibular disorder or a central brainstem or cerebellar lesion, and even outperform early neuroimaging in identifying dangerous brainstem strokes, these subtle eye movements often go unrecognized by front-line providers without specialized training in neuroophthalmology or neurootology (Wagle et al., 2022b). ENG (electrodes) and VNG (infrared goggles) offer quantitative nystagmus metrics (e.g., slow-phase velocity via caloric tests) but are costly, bulky, and confined to specialist centers. Consequently,

most rely on subjective bedside exams that often miss subtle nystagmus, driving demand for portable, accurate detection methods.

Recent advances in computer vision and deep learning offer a promising path toward automated, expert-level detection of nystagmus from standard video recordings (Punuganti et al., 2019; Phillips et al., 2019; Newman et al., 2021a,b; Reinhardt et al., 2020; Wagle et al., 2022a; Zhang et al., 2021a; Lu et al., 2022). Deep learning models can now replicate what clinicians observe segmenting pupils, tracking eye trajectories, classifying nystagmus types, and computing metrics like slow-phase velocity-to produce quantitative, clinician-friendly reports. These technologies open the door to low-cost, portable, and telehealth-compatible solutions, enabling patients to record eye movements on a smartphone and receive automated assessments remotely.

Despite this progress, key challenges remain. Main among them is the lack of large, high-quality training datasets for nystagmus detection in video (Lohr et al., 2020; Lohr and Komogortsev, 2022; Zola Matuvanga et al., 2021). Even when video-oculography data are available in research or specialty clinics, privacy and regulatory concerns often hinder data sharing, as videos may contain identifiable features and sensitive biometric health information. This scarcity of open datasets constrains the development and generalizability of deep learning models, which are often trained on limited, single-center data that may not perform reliably across diverse patient populations or real-world conditions.

To overcome the data bottleneck, researchers have started exploring synthetic data generation for nystagmus (Guibas et al., 2017; Garcea et al., 2023; Wang et al., 2023a). Emerging work in video generation models suggests that it is possible to create realistic eye movement videos that exhibit specific nystagmus waveforms, without using any real patient video in the training process (Rahman et al., 2025). Such controllable video generation allows tuning parameters like the nystagmus direction, amplitude, and frequency, producing a wide range of scenarios to train robust models. By leveraging open-source simulation data or procedurally generated eye movement videos, deep learning models could be trained on diverse "virtual" patients, making them more generalizable. In addition, synthetic data sidesteps privacy issues, enabling data sharing and collaborative research without risking confidential patient information.

To address the limitations in data availability and support generalizable, privacy-preserving nystagmus detection, we propose a novel framework that leverages synthetic waveform modeling and video diffusion transformers (Illustrated in Figure **??**). Our approach begins by mathematically modeling diverse synthetic nystagmus waveforms, incorporating variations in direction, amplitude, frequency, and noise to simulate real-world conditions. These waveforms are first validated through a deep learning-based waveform classifier trained on synthetic data and tested on waveforms from real-world nystagmus patient data.

To generate realistic eye movement videos, we condition a video diffusion model on synthetic pupil motion trajectories inspired by real-world video-oculography recordings. However, due to GPU memory constraints and the challenge of generating clinically viable long-form videos, the model alone cannot generate smooth, fine-grained nystagmus dynamics. To overcome this, we introduce a two-step generation pipeline: pupil segmentation masks are first generated from synthetic waveforms and then refined using a flow-based interpolation model to improve temporal consistency and clinical realism. Finally, we extract the waveforms

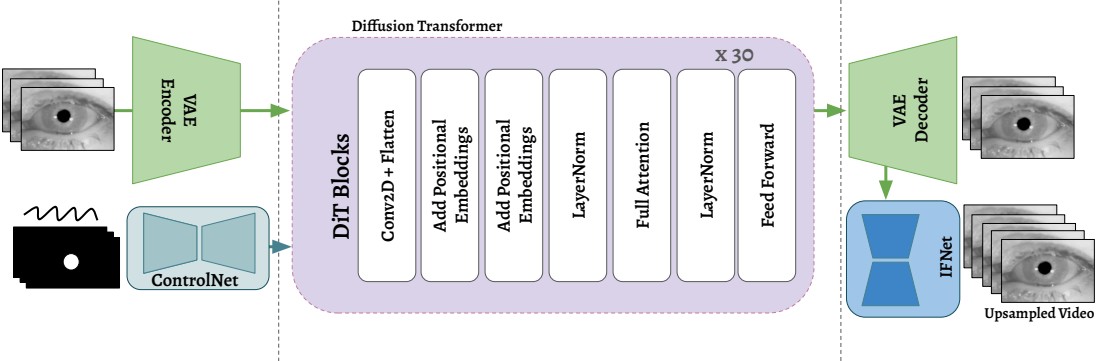

Figure 1: **Architecture of our pupil-conditioned video diffusion transformer.** The model adds a lightweight ControlNet to CogVideoX-2B, using CNN-encoded pupil masks to guide generation at every transformer layer. Real masks supervise training, synthetic masks drive inference. A decoder reconstructs frames, and IFNet (Huang et al., 2022) upscales temporal resolution at inference only.

from generated videos and quantitatively compare them against real patient data for validation. The contributions are summarized as:

- *Synthetic waveform generation:* We propose a mathematical modeling pipeline to generate diverse, realistic nystagmus waveforms with controllable parameters (e.g., noise, amplitude, frequency).

- *Cross-domain waveform validation:* A waveform classifier trained solely on synthetic waveforms is evaluated on real patient data, demonstrating the clinical relevance of our synthetic dataset.

- *Pupil-conditioned video diffusion:* We adapt a video diffusion model conditioned on real-world pupil trajectories to generate synthetic nystagmus videos.

- *Flow-based waveform refinement:* To address the coarse nature of segmentation mask outputs, we integrate a flow-based model to interpolate and refine pupil motion, enhancing temporal smoothness and clinical fidelity.

- *End-to-end waveform evaluation:* We extract and analyze waveform characteristics from generated videos and benchmark them against real patient waveforms to assess the realism and diagnostic utility of our approach.

This framework enables privacy-preserving, scalable nystagmus video synthesis, paving the way for robust, generalizable deep learning models in neuro-ophthalmology.

## 2. Modeling Synthetic Nystagmus Waveforms

To capture the diverse dynamics of nystagmus, we model the horizontal or vertical displacement of the pupil, $P(t)$, as a function of time, $t$, using a set of parametric equations. In

all models below, $A$ denotes the peak excursion (amplitude), $\omega$ the angular frequency of oscillation, and $\phi$ a phase offset that aligns the waveform with its initial condition. More details at Appendix B.

## 3. Controllable Video Diffusion Transformer

We build on the pretrained CogVideoX-2b text–to–video diffusion model by adding a lightweight ControlNet (Chen et al., 2023) branch for spatial conditioning (shown in the Figure 1). In the following section, we denote $x_0 \in \mathbb{R}^{T \times H \times W \times 3}$ be a ground-truth video, $p$ be its text prompt embedding, $c$ be a control map of shape $T \times H \times W$ and $\epsilon_\theta(\cdot)$ be the model's noise predictor. For the video diffusion transformer, we encode $x_0$ via a 3D VAE:

$$z_0 = \mathcal{E}(x_0) \in \mathbb{R}^{L \times d}, \quad L = T'H'W', \ d = \text{latent dim.}$$

The expert transformer then iteratively denoises:

$$z_t = z_{t-1} + \text{TransformerBlock}\big(z_{t-1},\, p\big), \quad t = 1, \ldots, T.$$

For controllable generation, a small CNN encoder $\Phi$ maps $c$ into a feature tensor $C \in \mathbb{R}^{L \times d}$. For each transformer layer $l$, we add a zero-initialized injection $W^{(l)}$:

$$C = \Phi(c), \tag{1}$$

$$\tilde{z}^{(l)} = \text{Block}^{(l)}\big(z^{(l-1)},\, p\big), \tag{2}$$

$$z^{(l)} = \tilde{z}^{(l)} \ + \ W^{(l)}(C). \tag{3}$$

Since $W^{(l)}$ starts at zero, the pretrained behavior is preserved until fine-tuning. We train only $\{\Phi,\, W^{(l)}\}$ by minimizing the standard DDPM loss with classifier-free guidance on both $p$ and $c$:

$$\mathcal{L} = \mathbb{E}_{t,\, x_0,\, \epsilon} \Big\| \epsilon - \epsilon_\theta\big(z_t,\, p,\, C,\, t\big) \Big\|^2, \tag{4}$$

$$z_t = \sqrt{\bar{\alpha}_t}\, z_0 + \sqrt{1 - \bar{\alpha}_t}\, \epsilon, \quad \epsilon \sim \mathcal{N}(0, I). \tag{5}$$

## 4. Synthetic Waveform-Guided Inference

We now describe how to convert a 1D pupil waveform into a sequence of spatial masks for controllable video inference. Let

$$w = \{w_i\}_{i=1}^{N}, \quad w_i \in \mathbb{R}$$

be the sampled pupil positions over time. Our transformer can generate at most $T' = 41$ frames, so we first coarse-sample $w$ at $T'$ indices:

$$S = \big\lfloor \tfrac{(i-1)\, N}{T'-1} \big\rfloor + 1, \quad \tilde{w}_i = w_{S_i}, \quad i = 1, \ldots, T'.$$

Each $\hat{w}_i$ is converted into a binary mask $c_i \in \{0,1\}^{H \times W}$ by placing a disk of radius

$$r_i = \gamma\, \hat{w}_i$$

centered at $(u_0, v_0)$ (the pupil centroid):

$$c_i(x, y) = \begin{cases} 1, & (x - u_0)^2 + (y - v_0)^2 \leq r_i^2, \\ 0, & \text{otherwise.} \end{cases}$$

Stacking these yields the control map

$$c = \begin{bmatrix} c_1, c_2, \ldots, c_{T'} \end{bmatrix} \in \{0, 1\}^{T' \times H \times W}.$$

During inference, we initialize $z_T$ with Gaussian noise and run the conditioned reverse diffusion:

$$z_{t-1} = z_t - \epsilon_\theta(z_t, p, \Phi(c), t) \Delta t, \quad t = T, \ldots, 1,$$

then decode:

$$\hat{x} = \mathcal{D}(z_0) \in \mathbb{R}^{T' \times H \times W \times 3}.$$

This process produces a $T'$-frame video whose pupil motion follows the original waveform $w$ via the spatial masks $c$. To recover a full-rate video of length $N \gg T'$ from the coarse $T'$-frame output $\{\hat{x}_i\}_{i=1}^{T'}$, we apply a real-time flow estimator (Huang et al., 2022) $\mathcal{F}$:

**Pairwise flow estimation.** For each adjacent frame pair $(\hat{x}_i, \hat{x}_{i+1})$, compute bidirectional flows

$$f_{i \to i+1} = \mathcal{F}(\hat{x}_i, \hat{x}_{i+1}), \quad f_{i+1 \to i} = \mathcal{F}(\hat{x}_{i+1}, \hat{x}_i).$$

**Temporal interpolation.** For any intermediate timestep $\tau \in (0, 1)$ between frames $i$ and $i + 1$, warp and blend:

$$\tilde{x}_i(\tau) = (1 - \tau) \mathcal{W}(\hat{x}_i, \tau f_{i \to i+1}) + \tau \mathcal{W}(\hat{x}_{i+1}, -(1 - \tau) f_{i+1 \to i}),$$

where $\mathcal{W}(\cdot, \cdot)$ denotes differentiable backward warping. By sampling $\tau = \frac{k}{M+1}$ for $k = 1, \ldots, M$ (with $M = \frac{N}{T'} - 1$), we reconstruct the complete sequence.

$$\left\{ \hat{x}_1, \tilde{x}_1(\tfrac{1}{M+1}), \ldots, \hat{x}_2, \ldots, \hat{x}_{T'} \right\}$$

of length $N$. Finally, we apply a 1D low-pass filter along the time axis to the upsampled video frames to eliminate any residual flicker while preserving fine motion details. The complete workflow is detailed in Algorithms 1 and 2.

## 5. Experiment

**Datasets.** We train our conditional generative model of the ocular region using videos from the Labeled Pupils in the Wild (LPW) dataset (Tonsen et al., 2016), which includes 66 high-resolution high-frame rate videos centered on the eye region, originally developed for pupil detection tasks. For supervised training, we also incorporate a private Nystagmus dataset as described in (Kocak et al., 2021). A separate validation set is constructed using videos from 5 normal and 5 nystagmus patients. The final training and test sets consist of 1000 and 200 short video clips, respectively.

**Synthetic Waveform Validation Model.** For our waveform validation classification experiments, we employ a four-stage convolutional encoder followed by a bidirectional LSTM

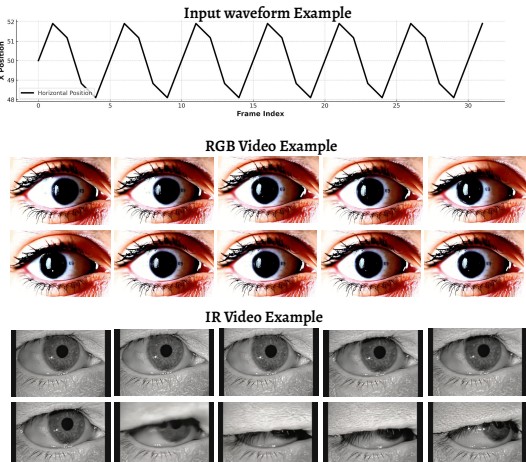

Figure 2: **Qualitative examples of the generated video frames.** For each case, we show the corresponding input waveform that guides the synthesis process, followed by representative RGB frames from generated eye videos and infrared (IR) frames from IR video outputs. These examples illustrate the model's ability to produce temporally coherent and anatomically consistent eye movements across both RGB and IR modalities.

with an attention-pooling head. Concretely, the raw input sequence ($B \times T \times F$) is first permuted to ($B \times F \times T$) and passed through four sequential 1D Conv-ReLU-BatchNorm-MaxPool blocks with channel widths $[32, 64, 128, 256]$, kernel size 5 and stride 1. The resulting feature map is then transposed back to ($B \times T \times 256$) and fed into a two-layer, bidirectional LSTM (hidden size $H$, dropout 0.5), whose full-sequence outputs are scored by a learned linear attention layer. The softmax-normalized attention weights are applied to the LSTM outputs to produce a single, $2H$–dimensional representation per example, which is layer-normalized and passed through an $H \rightarrow 4H$ linear projection (ReLU+dropout 0.5) and a final $H \rightarrow C$ classifier to yield the $C$ output logits.

**Baselines and Results.** Figure 2 shows qualitative results of the generated video frames, including representative RGB frames from synthesized eye videos and infrared (IR) frames from IR outputs. These examples highlight the model's ability to generate temporally coherent and anatomically consistent eye movements across both modalities. Table 1 presents the quantitative results alongside corresponding baselines. Each stage of the framework has its own baseline. For the *Waveform Classifier*, the baseline refers to a model trained on real patient waveform data and evaluated on the validation set. Our results represent the classifier's performance on synthetic data during inference to assess whether the model can accurately identify synthetic waveforms, thereby validating their realism. The accuracy decreases from 97.0% to 92.3%, and the Macro-F1 score drops from 96.1% to 90.1%, reflecting a modest reduction in performance when using synthetic data, but still supporting the realism of the generated waveforms. For *Video Generation* we

---

**Algorithm 1** Synthetic Waveform–Guided Inference

---

**Input:** 1D waveform $w = \{w_i\}_{i=1}^N$, transformer $D_\theta$, decoder $\mathcal{D}$, parameters $T'$, $\gamma$, pupil center $(u_0, v_0)$, noise schedule $\{\alpha_t\}$

**Output:** Generated coarse video frames $\{\hat{x}_i\}_{i=1}^{T'}$

**COARSE SAMPLING:**
  **for** $i \leftarrow 1$ **to** $T'$ **do**
    $S_i \leftarrow \lfloor \frac{(i-1)N}{T'-1} \rfloor + 1$   $\tilde{w}_i \leftarrow w_{S_i}$   $r_i \leftarrow \gamma \tilde{w}_i$   Generate mask $c_i(x, y) \leftarrow \mathbb{I}\{(x-u_0)^2 + (y-v_0)^2 \leq r_i^2\}$
  **end**

**STACK** $c \leftarrow [c_1, \ldots, c_{T'}]$

**DIFFUSION INFERENCE:**
  Sample $z_T \sim \mathcal{N}(0, I)$   **for** $t \leftarrow T$ **downto** 1 **do**
    $z_{t-1} \leftarrow z_t - D_\theta(z_t, p, \Phi(c), t) \Delta t$
  **end**

**DECODE:**
  **for** $i \leftarrow 1$ **to** $T'$ **do**
    $\hat{x}_i \leftarrow \mathcal{D}(z_0[i])$
  **end**

**return** $\{\hat{x}_i\}_{i=1}^{T'}$

---

**Algorithm 2** Flow-Based Video Upsampling & Temporal Smoothing

---

**Input:** Coarse frames $\{\hat{x}_i\}_{i=1}^{T'}$, flow estimator $\mathcal{F}$, upsample factor $M$, warping op. $\mathcal{W}$, smoothing filter $\mathcal{S}$

**Output:** Full-rate frames $\{x_t\}_{t=1}^N$, $N = T'(M+1)$

**INITIALIZE** empty list $\mathcal{V}$

**for** $i \leftarrow 1$ **to** $T' - 1$ **do**
  Append $\hat{x}_i$ to $\mathcal{V}$   $f_{i \rightarrow i+1} \leftarrow \mathcal{F}(\hat{x}_i, \hat{x}_{i+1})$   $f_{i+1 \rightarrow i} \leftarrow \mathcal{F}(\hat{x}_{i+1}, \hat{x}_i)$   **for** $k \leftarrow 1$ **to** $M$ **do**
    $\tau \leftarrow \frac{k}{M+1}$   $\tilde{x} \leftarrow (1-\tau)\mathcal{W}(\hat{x}_i, \tau f_{i \rightarrow i+1}) \quad + \tau \mathcal{W}(\hat{x}_{i+1}, -(1-\tau)f_{i+1 \rightarrow i})$   Append $\tilde{x}$ to $\mathcal{V}$
  **end**
**end**

Append $\hat{x}_{T'}$ to $\mathcal{V}$

**TEMPORAL SMOOTHING:**
  $\{x_t\} \leftarrow \mathcal{S}(\mathcal{V})$

**return** $\{x_t\}_{t=1}^N$

---

compare our method against GenVOG (Rahman et al., 2025), a training-free framework for nystagmus video generation that leverages a pretrained UNet-based architecture. To evaluate the visual quality and temporal coherence of the generated videos, we report results using following widely adopted metrics: Frechet Video Distance (FVD) (Unterthiner et al., 2019), which captures distribution-level similarity in temporal dynamics, and LPIPS, which assesses perceptual similarity at the frame level. Additionally, we also report Dynamic Degree, Imaging Quality and Motion Smoothness from VBench evaluation suite (Huang et al., 2024). We observe a lower LPIPS score (0.082 vs. 0.120), indicating higher perceptual

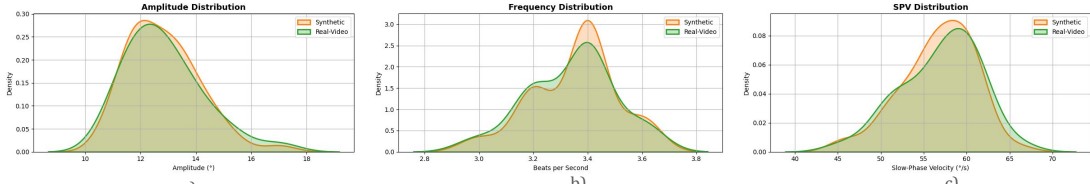

Figure 3: **Distribution shift analysis across waveforms.** We compare the distribution of three clinically relevant nystagmus waveform parameters: (a) amplitude, (b) frequency, and (c) slow-phase velocity (SPV): procedurally generated synthetic waveforms (orange) and waveforms extracted from real videos (green). The goal is to assess how well the synthetic signals, both direct and video-derived, match the statistical characteristics of real-world recordings. While synthetic waveforms closely resemble the real distributions in most metrics, minor shifts in SPV and frequency distributions suggest areas for further refinement.

similarity, and a substantial improvement in FVD (395 vs. 678), suggesting better temporal consistency in the generated videos. Then, for *Detection AUROC*, we train a classifier using three configurations: real data only, synthetic data only, and a combination of real and synthetic data. Classifier trained on synthetic data alone achieves an AUROC of 0.69. When trained on a combination of real and synthetic data, the performance improves to 0.92, slightly surpassing the classifier trained only on real data, which scored 0.89. This suggests that synthetic data can complement real data and enhance classifier performance. Finally, we conduct an ablation study by comparing the FVD scores of videos generated with and without the flow-based interpolation module. Removing the module results in a slightly higher FVD score (399 compared to 395), indicating degradation in video quality without the flow-based refinement step. Additionally, to assess the realism of our synthetic waveforms, we compare key clinical parameters: amplitude, frequency, and slow-phase velocity, across real patient recordings, procedurally generated synthetic signals, and waveforms extracted from generated videos (Figure 3). The distributions demonstrate strong alignment, particularly in amplitude, supporting the validity of our synthetic data for training and evaluation.

## 6. Related Work

**Video Generation Networks.** In recent years, there has been rapid progress in generative video modeling, with most state-of-the-art systems built on either denoising U-Net architectures (Blattmann et al., 2023; Singer et al., 2022; Ho et al., 2022a; Hong et al., 2022; Ho et al., 2022b; Mei and Patel, 2023; Molad et al., 2023; Wang et al., 2023b) or transformer-based diffusion backbones (Yang et al., 2024; Liu et al., 2024). These methods typically integrate spatial diffusion modules with temporal processing layers to jointly model frame-wise appearance and motion dynamics. Despite impressive results, two major challenges persist: (1) most high-quality video diffusion models remain closed-source, and (2) training them from scratch requires massive computational resources and large-scale video datasets.

| Exp. | Metric | Baseline | Ours | Δ |
|---|---|---|---|---|
| Waveform Classifier | Accuracy (%) | 97.0 | 92.3 | -4.7 |
| | Macro-F1 (%) | 96.1 | 90.1 | -6.0 |
| Video Generation | LPIPS ↓ | 0.120 | 0.082 | −0.038 |
| | FVD ↓ | 678 | 395 | −283 |
| | Dynamic Degree ↑ | 91.85 | 99.17 | +7.32 |
| | Motion Smoothness ↑ | 95.67 | 98.50 | +2.83 |
| | Imaging Quality ↑ | 63.17 | 70.5 | +7.33 |
| Detection AUROC | Real only | 0.89 | — | — |
| | Synthetic only | — | 0.69 | — |
| | Synth+Real | — | 0.92 | — |
| Ablation (w/o flow) | FVD ↓ | 399 | 395 | −4 |
| | Motion Smoothness ↑ | 96.1 | 98.50 | +2.4 |

Table 1: **Quantitative results across all experimental settings.** The column labeled "Baseline" corresponds to existing methods or specific ablated variants, while "Ours" refers to the performance of our full proposed pipeline. Metrics are chosen to reflect both perceptual and temporal qualities across tasks. Downward arrows (↓) indicate that lower values are preferred, such as in LPIPS and FVD, where lower scores denote better perceptual similarity and temporal coherence. These results collectively demonstrate the effectiveness of our approach in waveform classification, video generation, detection robustness, and the contribution of each component via ablation.

**Deep learning for Nystagmus Modeling.** One of the early efforts was the development of diagnostic decision support systems using CNN-based models for benign paroxysmal positional vertigo (BPPV) detection, achieving high sensitivity and specificity across horizontal, vertical, and torsional directions (Lim et al., 2019b). Later work introduced aEYE, a deep learning model trained to detect nystagmus beats in short videos with an AUROC of 0.86 and an accuracy of 82.7%, using simple 1D CNN architectures on labeled video clips (Wagle et al., 2022b). Specialized models for vertical and torsional nystagmus classification have also been proposed. The model in (Li and Yang, 2023d) achieved 91% accuracy in vertical nystagmus recognition, while torsional nystagmus classification with a transformer-based approach reached 92.9% test accuracy (Li and Yang, 2023c). The LAD hybrid system combined LSTM and CNN modules and achieved 91% accuracy on a large dataset of positional tests (Pham et al., 2022). Smartphone-based solutions have emerged for low-cost tracking. Con-VNG, a CNN-based system, was proposed for analyzing slow-phase velocity in smartphone videos (Friedrich et al., 2023). Similarly, EyePhone used smartphone cameras and showed strong correlation with infrared VOG for detecting optokinetic responses (Bastani et al., 2024). Another lightweight real-time nystagmus tracking framework based on ocular object segmentation demonstrated robust feasibility in clinical settings (Cho et al., 2024).

Generative and transformer-based architectures have also been explored. The GPT-4 Vision model was repurposed to classify nystagmus patterns but performed below expectation,

achieving only 37% accuracy overall (Noda et al., 2025). This highlights a gap in domain adaptation and the need for more targeted architectures. Lastly, telehealth frameworks integrating deep learning for nystagmus detection have shown promise in remote diagnostics. A deep learning-based system trained on 15,000 video frames achieved 98% accuracy, emphasizing the potential of large-scale, annotated video data for real-world deployment (Sanghvi et al., 2025).

Recent studies have applied machine learning to detect and analyze nystagmus using video-oculography (VOG) data, including both video and waveform signals (Punuganti et al., 2019; Phillips et al., 2019; Newman et al., 2021a,b; Reinhardt et al., 2020; Wagle et al., 2022a; Zhang et al., 2021a; Lu et al., 2022). However, these efforts are limited by the use of small, private datasets with few nystagmus variations, largely due to patient privacy concerns (Lohr et al., 2020; Lohr and Komogortsev, 2022; Zola Matuvanga et al., 2021), which hinders reproducibility and broader research. Modeling nystagmus dynamics remains a complex challenge due to the intricate interplay of ocular motor physiology and biomechanics influenced by eye, head, and body position (Punuganti et al., 2019; Phillips et al., 2019; Newman et al., 2021a,b; Reinhardt et al., 2020; Wagle et al., 2022a; Lu et al., 2022; Lim et al., 2019a; Zhang et al., 2021b; Li and Yang, 2023a,b). To mitigate dataset scarcity in medical applications, researchers often rely on synthetic data generation from real-world samples (Guibas et al., 2017; Garcea et al., 2023; Wang et al., 2023a). Yet, this approach can suffer from overfitting, where generative models reproduce training data too closely, defeating the goal of generalization (Somepalli et al., 2023). In the case of nystagmus, the challenge is amplified by the need to generate long-form, clinically meaningful videos that accurately reflect physiological pupil-position waveforms.

## 7. Conclusion

We have presented a novel, end-to-end framework for privacy-preserving nystagmus video synthesis and analysis. By combining (1) mathematically modeled synthetic waveforms, (2) a waveform classifier for cross-domain validation, (3) pupil-conditioned video diffusion, and (4) flow-based interpolation for temporal refinement, our pipeline generates high-fidelity eye-movement videos that faithfully reproduce clinically relevant nystagmus dynamics. Extensive experiments demonstrate that our synthetic datasets enable classifiers and regression models to generalize to real patient recordings, while sidestepping privacy and data-scarcity challenges inherent to clinical video-oculography.

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

## Appendix A. Limitations and Ethics Statement.

While our framework advances scalable, privacy-preserving nystagmus video synthesis, several limitations must be acknowledged. First, although synthetic waveforms are designed to mimic real patient dynamics, our generated videos may still lack subtle anatomical

or contextual cues present in clinical recordings (e.g., eyelid motion, head oscillation, lighting variability). Second, the diffusion model's coarse mask outputs and subsequent flow-based interpolation may introduce artifacts under extreme waveform parameters (very high frequency or amplitude), potentially misleading downstream classifiers. Additionally, due to the stochastic nature of diffusion models, certain random seeds can lead to unrealistic visual anomalies, such as duplicate pupils, excessive noise, or nonsensical motion artifacts, that may compromise video fidelity. Third, our current evaluation focuses primarily on horizontal jerk and pendular patterns; extension to vertical, torsional, or mixed-pattern nystagmus will require further modeling and validation. Finally, the computational cost of diffusion and optical-flow upsampling may limit real-time deployment on resource-constrained devices.

Ethically, synthetic data can mitigate privacy concerns by eliminating identifiable patient features, yet it may also foster overconfidence: models trained purely on artificial videos must be rigorously validated on diverse clinical datasets before clinical use. There is a risk of misuse if synthetic videos are mistaken for true patient recordings or used inappropriately (e.g., during forensic or insurance assessments). We therefore recommend that any diagnostic tool developed with our pipeline be integrated under expert supervision, with clear disclaimers about the synthetic nature of training data and strict adherence to medical device regulations and institutional review guidelines.

## Appendix B. Modeling Synthetic Nystagmus

Figure 5 presents examples of both synthetic and real waveforms. Figure 6 shows real-world example of pupil movements overtime for Nystagmus patient.

1. *Pendular Nystagmus.* Pendular nystagmus exhibits smooth, sinusoidal oscillations akin to a physical pendulum. We define

$$P_{\mathrm{pend}}(t) \;=\; A \, \sin\!\big(\omega\, t + \phi\big).$$

2. *Accelerating Jerk Nystagmus.* Here, the slow phase accelerates quadratically away from center before a rapid corrective saccade returns the gaze. One simple cycle-based approximation is

$$P_{\mathrm{jerk\_acc}}(t) \;=\; A\,\big(t^2 + \phi\big) \quad (0 \le t < T),$$

with a discontinuous reset at $t = T$.

3. *Decelerating Jerk Nystagmus.* In this variant, the slow-phase velocity decreases linearly over time, producing a concave trajectory:

$$P_{\mathrm{jerk\_dec}}(t) \;=\; A\,\big(t - 0.5\,t^2\big) \quad (0 \le t < T),$$

followed by a fast reset at the end of each period.

4. *Linear Jerk Nystagmus.* When the slow phase proceeds at constant velocity before a quick reversal, the motion is piecewise linear:

$$P_{\mathrm{jerk\_lin}}(t) = \begin{cases} A\,(-0.5\,t), & 0 \le t < T, \\ 0, & t = T, \end{cases}$$

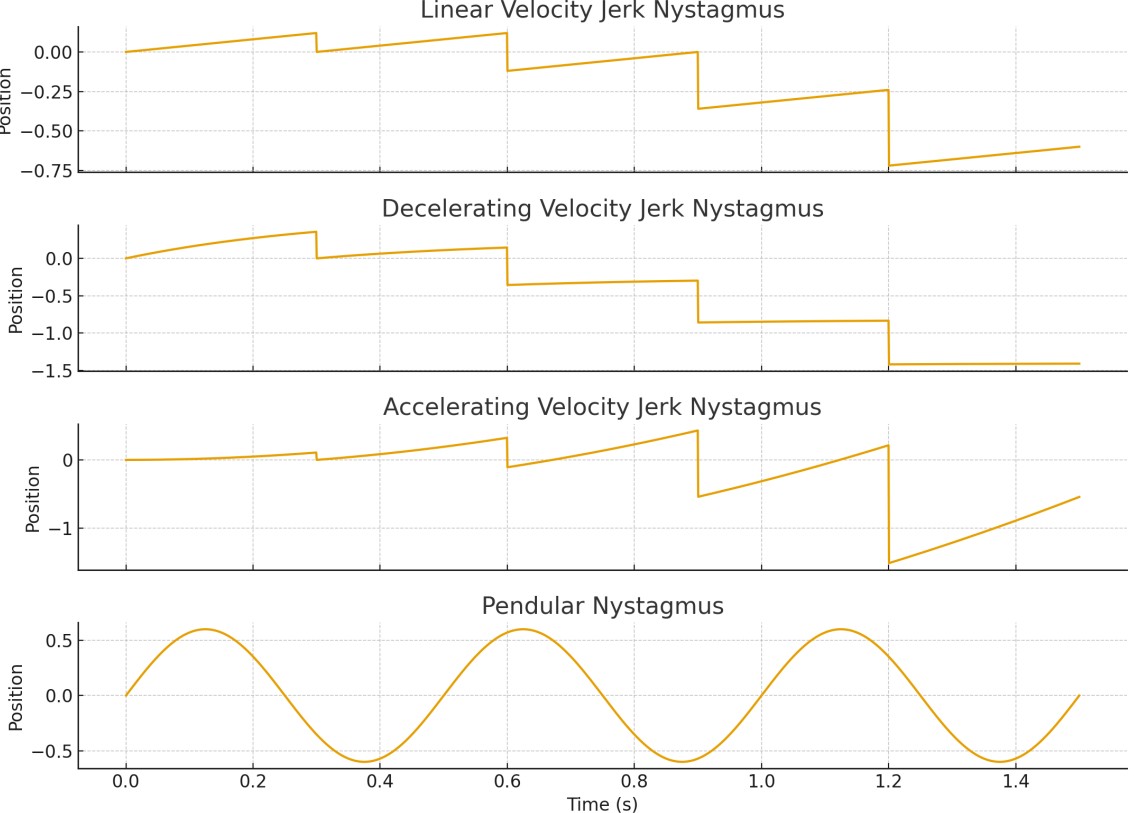

Figure 4: Waveform morphologies of various nystagmus types. Nystagmus can be categorized into two main types based on the pattern of eye movements: jerk nystagmus, which consists of a slow phase followed by a quick corrective phase, and pendular nystagmus, characterized by oscillations that are slow in both directions. These waveform patterns reflect abnormalities in the neural pathways responsible for maintaining gaze stability. Jerk nystagmus can be further subdivided according to the velocity profile of the slow phase, including linear velocity, decelerating velocity, and accelerating velocity waveforms.

then repeats every $T$.

5. *Square-Wave Jerk.* Square-wave jerks consist of brief, step-like deviations from fixation followed by corrective steps. We model this as

$$P_{\text{sq}}(t) = \begin{cases} +A, & 0 \leq t < \frac{T}{2}, \\ -A, & \frac{T}{2} \leq t < T, \end{cases}$$

with periodic repetition.

Figure 4 illustrates different Nystagmus types.

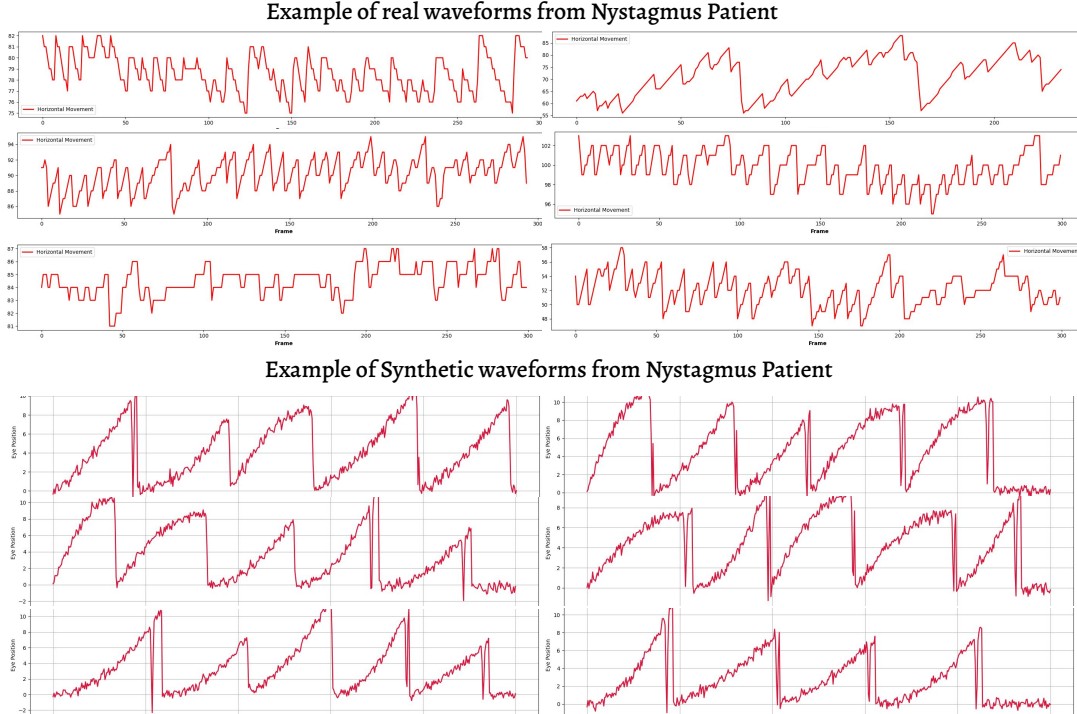

Example of real waveforms from Nystagmus Patient

Example of Synthetic waveforms from Nystagmus Patient

Figure 5: **The figure illustrates a comparison between real and synthetic nystagmus waveforms.** We focus on jerk nystagmus, the only subtype present in our real patient dataset. For comparison, we present a real patient waveform recorded over 300 frames alongside a synthetically generated waveform rendered over a 5-second interval. While only jerk nystagmus is shown here due to limitations in real data availability, our generative framework is capable of realistically simulating a broad range of nystagmus patterns, including pendular, vertical, and mixed-type oscillations.

**Parameter Selection.** Following Kocak et al. (Kocak et al., 2021), we constrain the slow-phase velocity by specifying an initial velocity $v_0 \in [5, 25]\,°/s$ (in 0.5°/s steps) and a velocity decay of 0-24% over each cycle. These bounds ensure our synthetic waveforms span clinically observed ranges.

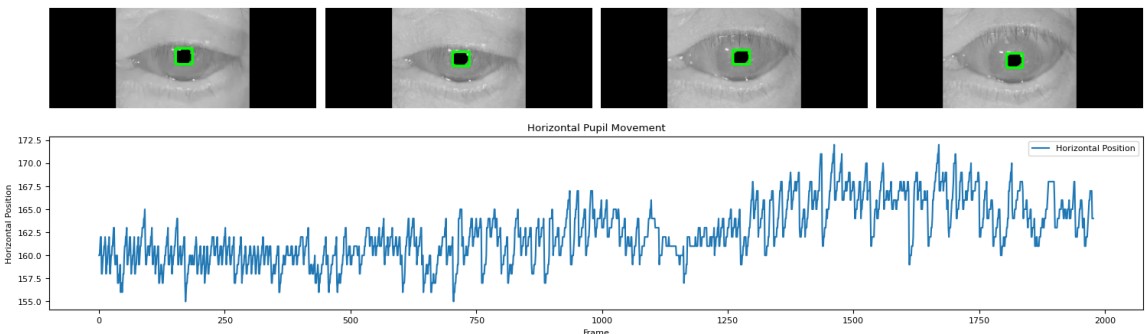

Figure 6: Real-world example of Nystagmus by tracking pupil.

