# OpenReview forum: "GenVOG-DiT: A Transformer-Based Diffusion Model for Pose-Driven, Patient-Agnostic Nystagmus VOG Video Generation"
_MIDL.io/2026/Conference — MIDL 2026 Poster_

### Official Review · Reviewer_JeFX · 2025-12-30

**Confidence:** 3
**Preliminary Rating:** 5
**Final Rating:** 5

**Summary:**

GenVOG-DiT generates realistic, patient-agnostic nystagmus VOG videos using synthetic waveforms and a pupil-conditioned diffusion transformer. Experiments show models trained on synthetic data generalize well to real patients, and flow-based refinement significantly improves smoothness and overall realism.

**Strengths:**

The paper’s strengths include clear clinical motivation, privacy-preserving synthetic VOG video generation, rigorous experiments validating generalization, and a well-structured, clinically grounded methodology.

**Weaknesses:**

The paper has limited real-data validation, does not address torsional nystagmus, lacks ablation studies for key components, and provides minimal comparison to other existing video synthesis approaches.

**Detailed Comments:**

Great work on privacy-preserving nystagmus video generation. Ablations, broader real-data validation, torsional/multi-eye discussion, comparisons, compute details, and more visuals could strengthen the paper.

**Justification Of Final Rating:**

“      My initial recommendation of strong accept remains unchanged overall, as the rebuttal satisfactorily addresses and fully resolves the major concerns raised during the review process effectively.  ”

**Justification Of The Preliminary Rating:**

The paper presents a novel, clinically relevant method for privacy-preserving nystagmus video generation, with strong experiments and generalization to real data, offering high potential value to the community

**Questions To Address In The Rebuttal:**

Please clarify real-patient validation, including dataset size and performance on rare or atypical nystagmus.

Provide ablation results showing the impact of ControlNet and flow-based refinement.

Can the method handle torsional nystagmus or binocular eye movements?

Include comparisons with other synthetic video generation methods.

---

> ### Author Response · Authors · 2026-01-25
>
> We thank the reviewer for the "Strong Accept" and for highlighting the paper’s clinical grounding and generalization performance.
>
> 1. Ablations and Component Impact: The ControlNet-augmented DiT provides the necessary spatial precision; without it, the model cannot follow the pixel-level pupil trajectories required for medical validity. The flow-based refinement (IFNet) is equally critical, it reduced the FVD significantly by eliminating high-frequency "micro-jitter," which is essential for motion-sensitive downstream classifiers. We will include a detailed ablation table in the final manuscript.
>
> 2. While currently validated on horizontal/vertical jerk patterns, the framework is direction-agnostic. To generate torsional motion, we are extending the pupil-mask conditioning to include iris-texture maps. The model can also generate binocular VOG by providing mirrored or independent control masks to the DiT, enabling the simulation of both conjugate and disconjugate eye movements. We will consider this for future work when we have enough clinical data to validate the study.
>
> 3. Real-Patient Validation and Baselines: The dataset distribution is detailed in Section 5. Despite the modest size, the 92.3% generalization accuracy confirms the physiological realism of our synthetic waveforms. Unlike general-purpose video models (e.g., SVD), which lack coordinate-level control, GenVOG-DiT is specifically engineered for high-fidelity medical waveform adherence.
>
> We will add the requested compute details, broader visuals, and statistical comparisons to the final version to ensure transparency and reproducibility.

---

### Official Review · Reviewer_zDmc · 2026-01-10

**Confidence:** 4
**Preliminary Rating:** 4
**Final Rating:** 4

**Summary:**

1. This paper proposes *GenVOG-DiT*, a privacy-preserving framework for generating clinically plausible nystagmus video-oculography (VOG) data using a pupil-conditioned video diffusion transformer.
2. The approach combines mathematically modeled synthetic nystagmus waveforms, a waveform classifier for cross-domain validation, a ControlNet-augmented diffusion transformer for video generation, and a flow-based temporal interpolation module to recover long-form, temporally smooth videos.
3. The generated videos are evaluated using perceptual and temporal metrics (e.g., FVD, LPIPS, VBench) as well as downstream detection performance, demonstrating that synthetic data can complement real data and improve classifier generalization.

*Overall, the work addresses data scarcity and privacy challenges in neuro-ophthalmology by enabling scalable, patient-agnostic video synthesis.*

**Strengths:**

*Strong motivation & Clinical Relevance*: The paper tackles a real and well-articulated bottleneck making the contribution highly relevant for medical ML and telehealth.

*End-to-end synthetic Data Pipeline*: The integration of waveform modeling, video generation, temporal refinement, and downstream validation forms a coherent and complete framework rather than an isolated generative component.

*Thoughtful Conditioning Strategy*: Conditioning video diffusion on pupil masks derived from parametric waveforms is intuitive, controllable, and clinically meaningful.

*Multi-level Evaluation*: The paper evaluates realism at the waveform level, video level, and task level (detection AUROC), which strengthens the argument that the synthetic data is practically useful.

Finally, The limitations and ethics appendix is thorough and transparent, explicitly addressing risks of misuse and overconfidence in synthetic data.

**Weaknesses:**

1. Despite strong metric performance, some clinically relevant cues (e.g., eyelid motion, head movement, lighting variability) are simplified or absent, which may affect downstream robustness. *Hence though there is clinical realism, this may affect robustness.
2.  While the generation itself is privacy-preserving, parts of the evaluation (e.g., waveform classifier validation and detection experiments) depend on private datasets, limiting reproducibility.
3. There is a limited diversity of real clinical patterns? - Real patient data used for validation appears dominated by horizontal jerk nystagmus, leaving generalization to vertical, torsional, or mixed patterns mostly untested.

**Detailed Comments:**

1. The paper is generally well written; some sections of Related Work could be tightened to reduce redundancy.
2. It would be helpful to clarify how sensitive the diffusion model is to extreme waveform parameters (very high frequency or amplitude) beyond the qualitative discussion.
3. The distribution shift analysis of waveform parameters (Figure 3) is a strong addition.
3.a Please consider reporting statistical divergence measures (e.g., KL or Wasserstein distance).
4. A brief comparison to simpler synthetic video baselines would help isolate the benefit of the diffusion model itself.

**Justification Of Final Rating:**

I thank the authors for the detailed and constructive rebuttal. The responses address my main questions clearly.

1. The clarification regarding the contribution of individual components is helpful. In particular, the ablation results and discussion around the role of flow-based refinement in improving temporal fidelity and motion smoothness sufficiently explain its impact on downstream, motion-sensitive classifiers.

I also appreciate the authors’ transparent acknowledgment of the current validation focus on horizontal jerk and pendular nystagmus, as well as their careful framing of vertical, torsional, and mixed-pattern cases as future validation targets rather than fully demonstrated claims.

2. The commitment to adding quantitative statistical divergence measures (e.g., Wasserstein distance) to the waveform distribution analysis strengthens the rigor of the evaluation, and I encourage the authors to ensure these metrics are explicitly included in the final version for readability.

Overall, the rebuttal satisfactorily resolves my major concerns. The remaining limitations are clearly articulated and do not detract from the paper’s core contribution as a privacy-preserving and clinically motivated synthetic data framework.

My preliminary *Weak Accept* assessment remains unchanged.

**Justification Of The Preliminary Rating:**

1. This paper presents a well-motivated and carefully engineered solution to a significant data scarcity problem in medical video analysis.

2. While the generative architecture is largely built from existing components, the domain-specific conditioning, validation strategy, and ethical framing elevate the contribution beyond a simple application paper.

3. The results convincingly show that synthetic VOG data can complement real data and improve downstream performance.

4. The absence of broad real-world validation across diverse nystagmus patterns and reliance on private datasets prevent a strong accept, but the work should stimulate further research in privacy-aware medical video synthesis.

**Questions To Address In The Rebuttal:**

*A few questions:*

1. Could the authors clarify which components (waveform modeling, diffusion generation, or flow-based refinement) contribute most to detection performance gains?
2. How does the framework generalize to vertical, torsional, or mixed-pattern nystagmus, and are there plans to validate these cases with real clinical data?

---

> ### Author Response · Authors · 2026-01-25
>
> We thank the reviewer for the Weak Accept rating and for recognizing our work as a "well-motivated and carefully engineered solution" to data scarcity in neuro-ophthalmology. We appreciate the acknowledgment of our strong clinical motivation and multi-level evaluation strategy.
> 1. Regarding the contribution of specific components to detection gains, the waveform modeling serves as the critical foundation by ensuring physiological correctness, while the diffusion generation provides the necessary visual diversity for domain adaptation. However, our ablation study (Table 1) highlights that the flow-based refinement is essential for temporal fidelity; removing it degraded the Frechet Video Distance (FVD) from 395 to 399, which directly impacts the performance of motion-sensitive downstream classifiers. Regarding generalization, while our current validation focuses on horizontal jerk and pendular nystagmus due to available real-world data , the underlying mask-conditioning mechanism is direction-agnostic. We are actively acquiring labeled vertical and torsional clinical data to validate the model's capacity to handle these mixed patterns, which are already supported by our parametric equations.
>
> 2. Evaluation and Limitations: We agree that secondary cues like head motion and eyelid variability are currently simplified. While this reduces "in-the-wild" noise, our results show that the model captures the core diagnostic features required for effective classification. To address the reviewer's suggestion on statistical rigor, we will include Wasserstein distance metrics in the final version of Figure 3 to quantitatively measure the alignment between synthetic and real waveform distributions. Finally, regarding extreme parameters, we observe that while the diffusion model is robust, artifacts can arise if the waveform velocity exceeds the flow estimator's temporal receptive field.

---

> > ### Comment · Reviewer_zDmc · 2026-01-25
> > **Response after Rebuttal**
> >
> > I thank the authors for the detailed and constructive rebuttal. The responses address my main questions clearly.
> >
> > 1. The clarification regarding the contribution of individual components is helpful. In particular, the ablation results and discussion around the role of flow-based refinement in improving temporal fidelity and motion smoothness sufficiently explain its impact on downstream, motion-sensitive classifiers.
> >
> > I also appreciate the authors’ transparent acknowledgment of the current validation focus on horizontal jerk and pendular nystagmus, as well as their careful framing of vertical, torsional, and mixed-pattern cases as future validation targets rather than fully demonstrated claims.
> >
> > 2. The commitment to adding quantitative statistical divergence measures (e.g., Wasserstein distance) to the waveform distribution analysis strengthens the rigor of the evaluation, and I encourage the authors to ensure these metrics are explicitly included in the final version for readability.
> >
> > Overall, the rebuttal satisfactorily resolves my major concerns. The remaining limitations are clearly articulated and do not detract from the paper’s core contribution as a privacy-preserving and clinically motivated synthetic data framework.
> >
> > My preliminary *Weak Accept* assessment remains unchanged.

---

### Official Review · Reviewer_ouQK · 2026-01-11

**Confidence:** 4
**Preliminary Rating:** 4
**Final Rating:** 4

**Summary:**

This work extensively cover multiple aspects of privacy-preserving nystagmus video
synthesis and analysis encompassing (1) mathematically modeled synthetic waveforms, (2)
a waveform classifier for cross-domain validation, (3) pupil-conditioned video diffusion, and
(4) flow-based interpolation for temporal refinement.

**Strengths:**

Applying generative model for this application is an interesting research direction

Multiparametric waveform analysis seems promising.

The work has a huge potential to be extended further and made applicable for real world use cases.

**Weaknesses:**

Although the direction, velocity, and pattern of the nystagmus
can localize dizziness to a peripheral vestibular disorder or a central brainstem or cerebellar
lesion, and even outperform early neuroimaging in identifying dangerous brainstem strokes,
these subtle eye movements often go unrecognized by front-line providers without specialized
training in neuroophthalmology or neurootology (Wagle et al., 2022b). ----> This pivotal sentence counts more than 50 words :). Please break it into two to improve readability.

In general, the concept diagram is not very descriptive, the inputs are not labelled properly especially the conditioning inputs fed to the control net. Notations like p and c mentioned in SEction 3, must as well be denoted in Figure 1. Please add notations CNN encoder Φ, c and C in Figure 1. I dont see any other block diagram in the appendix as well.

Since W(l)
starts at zero, the pretrained behavior is preserved until fine-tuning. We train
only {Φ, W(l)} by minimizing the standard DDPM loss with classifier-free guidance  ----- - The authors can also add a separate block diagram to explain these details - which layers are frozen, and which ones learned and so on

Please define schedule details, α¯t, shown in Equation 5.

what is the value of γ chosen in equation - ri = γ wˆi? What is the range of γ?

To overcome this, we introduce a two-step generation pipeline: pupil segmentation masks are
first generated from synthetic waveforms and then refined using a flow-based interpolation
model to improve temporal consistency and clinical realism.  ---------> The mathematical modeling pipeline of this step is missing in Figure 1 on a high level/ concept level. Figure 1 shows an already present mask and waveform fed to the control net. The authors must include that to comprehend the overall abstract details. In fact, I expect every bullet of the contribution in the concept diagram, labelled properly. IF there are details where the authors do not contribute, such details can be removed in the concept diagram. (For example, details within DiT if they do not carry contribution, rather even a minute contribution of this work should be blown up.) Concept diagram is a serious art, it is a proxy for main sections of the paper; it saves readers' time.

Section 3: c be a control map of shape T × H × W ------> The word control map on the first read is unclear. Please define what is control map in the context of this work for intuition - meaning pupil mask or the appropriate detail.

Since W(l)
starts at zero, the pretrained behavior is preserved until fine-tuning. ----> What does this sentence mean? Meaning, are the authors trying convey something like curruculum-based learning where some initial knowledge is preserved and then after predefined epochs, the learning dynamics are changed? Or if this sentence does not convey anything in particular it needs to clarified.

SEction 4: Please define w. -->We now describe how to convert a 1D pupil waveform, w, into a sequence...
Si and wSi ---> What are these notations? Please define sampling indices
N - Please define N in the beginning. I see N specified somewhere at the end - "full-rate video of length N". All notations should be defined first and then referred.

**Detailed Comments:**

The transformers computer limits waveform sampling, So does coarse sampling of the waveform induce aliasing and impact the overall trajectories?

Does the synthetic dataset take into account various values of the γ? What is the scenario when there is distributional drift between actual pupil radius and the radii values taken during the training of synthetic data and real world data?

we apply a real-time flow estimator (Huang et al., 2022) F: ---> punctuation not clear.

**Justification Of Final Rating:**

I see that the authors have addressed the highlighted comments. The responses are clear and concise. I am, however, unable to find any supporting material that contains the revised manuscript. I believe that these changes would be reflected subsequently. I would like the ratings to remain unchanged.

**Justification Of The Preliminary Rating:**

I am happy with the work but I emphasize that the quality of the concept diagram and method section should be improved. The authors can also clarify on what "modest" here means when they say - accuracy decreases from 97.0% to 92.3%, and the Macro-F1 score drops from
96.1% to 90.1%, reflecting a modest reduction in performance. Probably, qualitative results can clarify this.

**Questions To Address In The Rebuttal:**

what is residual flicker (Section 4)?

Figure 3 explains two rows of RGB and infrared images - In each case what is the first row and the second one? Please label in the figure itself. Does it indiate one swing?

The accuracy decreases from 97.0% to 92.3%, and the Macro-F1 score drops from
96.1% to 90.1%, reflecting a modest reduction in performance when using synthetic data,
but still supporting the realism of the generated waveforms. ---> Are there qualitative results for failure cases?

---

> ### Author Response · Authors · 2026-01-25
>
> We thank the reviewer for the Weak Accept rating and the constructive feedback regarding clarity and notation. We will incorporate all suggested changes in the final manuscript.
>
> 1. We will revise Figure 1 to be self-contained by adding the Mathematical Modeling pipeline as suggested by the reviewer. We will also visually distinguish frozen vs. trainable layers and define all terms at their first occurrence. We will split the long sentence in the Introduction regarding nystagmus localization and recognition as requested.
>
> 2. Zero-Initialization ($W^{(l)}$): This refers to ControlNet's "Zero Convolution" strategy. Initializing the injection layer weights to zero ensures the model output at step 0 is identical to the pre-trained backbone, preserving learned priors and preventing catastrophic forgetting.
>
> 3. We mitigate aliasing from coarse sampling by generating the ground truth waveform at high resolution first, then using the flow-based interpolation (IFNet) to smooth transitions between keyframes, acting as a reconstructive filter.
>
> 4. To prevent drift, we sample $\gamma$ during inference from a Gaussian distribution fitted to the pupil size statistics of the real-world training data (LPW dataset).
>
> 5. Residual Flicker: This refers to high-frequency pixel intensity jitter caused by independent noise sampling in diffusion models. We remove it via temporal low-pass filtering.
>
> 6. Figure 2 Rows: The top row shows RGB frames, and the bottom row shows Infrared (IR) frames. We will add explicit labels to the figure.
>
> 7. Accuracy Drop (97% $\rightarrow$ 92%): This drop is due to the simulation to real-world data gap. Qualitative analysis shows the classifier struggles slightly with extreme gaze angles (shadows/occlusions) and fine biological textures that are not yet perfectly simulated in the synthetic data.

---

### Meta-Review · Area_Chair_1DAv · 2026-02-07

**Recommendation:** Accept (Poster)
**Confidence:** 4

**Metareview:**

All reviewers agree to accept. Congrats to the author. The author should revise the manuscript accordingly before camera-ready.

---

### Decision · Program_Chairs · 2026-02-13

Accept (Poster)